# A Novel Diamine Containing Ester and Diphenylethane Groups for Colorless Polyimide with a Low Dielectric Constant and Low Water Absorption

**DOI:** 10.3390/polym14214504

**Published:** 2022-10-25

**Authors:** Jun Seok Lee, Yong-Zhu Yan, Sung Soo Park, Suk-kyun Ahn, Chang-Sik Ha

**Affiliations:** 1Department of Polymer Science and Engineering, School of Chemical Engineering, Pusan National University, Busan 46241, Korea; 2Division of Advanced Materials Engineering, Dong-Eui University, Busan 47340, Korea

**Keywords:** polyimide, a novel diamine, low water absorption, low dielectric constant, high transparency, high solubility

## Abstract

In this study, a novel diamine monomer containing ester and phenyl moieties, 1,2-diphenylethane-1,2-diyl bis(4-aminobenzoate) (1,2-DPEDBA), was synthesized through a three-step reaction. Using this diamine, a novel polyimide (PI) film was prepared with 4,4′-(hexafluoroisopropylidene)diphthalic anhydride (6-FDA) as a counter dianhydride through a typical two-step chemical imidization. For comparison, poly(pyromellitic dianhydride-co-4,4′-oxydianiline) (PMDA-ODA PI) was also synthesized via thermal imidization. The resulting 6-FDA-DPEDBA PI film was not only soluble in common polar solvents with high boiling points, such as N,N-dimethylacetamide (DMAc) and N,N-dimethylformamide (DMF), but also soluble in common low-boiling-point polar solvents, such as chloroform (CHCl_3_) and dichloromethane (CH_2_Cl_2_), at room temperature. The resulting novel PI showed a 5% weight loss temperature (T^5^_d_) at 360 °C under a nitrogen atmosphere. The resulting PI film was colorless and transparent with a transmittance of 87.1% in the visible light region ranging from 400 to 760 nm. The water absorption of the novel PI film was of 1.78%. The PI film also possessed a good moisture barrier and hydrophobicity. Furthermore, the resulting PI film displayed a low dielectric constant of 2.17 at 10^6^ Hz at room temperature. In conclusion, the novel PI film exhibited much better optical transparency, lower moisture absorption, and a lower dielectric constant as well as better solubility than the PMDA-ODA PI film, which is insoluble in any solvent, although its thermal stability is not better than that of PMDA-ODA PI.

## 1. Introduction

With the fast growth of wireless mobile communication industries, the demand for the Internet of things (IoT), foldable displays, holograms, integrated circuits, and flexible substrates is also rapidly rising [1,2,3,4]. To meet these demands, the signal transmission rate and signal integrity should be high enough [5,6]. Additionally, for flexible displays and flexible devices, it is essential to have high transparency, high-temperature stability, low dielectric constant, low water absorption, and easy processing [7]. Traditional materials such as glass and metal foil cannot meet these requirements [8,9]. Many kinds of transparent or low dielectric constant polymers have been reported, for instance, polyimides [10,11,12,13], poly(ethylene naphthalate) (PEN) [14], polybenzimidazoles [15], polybenzoxazoles (PBO) [16], and poly(phenylene ether) (PPE) [17]. Among them, polyimide (PI) is the most promising material for next-generation mobile communication devices due to its outstanding high-temperature resistance, chemical resistance, mechanical properties, and tunable dielectric properties [18].

Aromatic PI has a charge transfer complex (CTC) between an electron-withdrawing dianhydride moiety and an electron-donating diamine moiety, and it interacts with different polymer chains, hindering their chain segmental mobility. Due to the CTC, PI possesses excellent high-temperature stability, high chemical resistance, and mechanical properties [19]. However, due to the CTC, PI cannot be dissolved into common organic solvents, has a high dielectric constant, and always shows deep colors and low transparency in the visible light region [20]. These kinds of disadvantages are quite fatal in the application of flexible displays or optoelectronic industries, in which transparency and processability are important [21]. Therefore, to satisfy the industrial requirements, modifying PI to obtain high-temperature stability, optical transparency, low dielectric properties, and good processability has been vastly studied. There are extensive studies on fabricating PI with a low dielectric constant and high transparency while maintaining the excellent features of PI [22,23,24,25,26]. To satisfy the industrial needs, many reports have focused on modifying the chemical structure of PI. These reports are usually focused on introducing fluorine-containing groups, bulky groups, asymmetric moieties, and flexible groups into the main chain of PI.

Kuo et al. [27] synthesized 36 different kinds of PI with various functional groups and related the dielectric properties with the different chemical structures. The researchers explained that structural parameters, such as the fluorine content (F%) and imide group content (Imide%), are strongly correlated with the dielectric constant and dielectric loss. Nam et al. [28] designed different kinds of poly(ester imide)s with various side groups and substituted positions. They mentioned that introducing ester moiety into the polymer chain can distort the main chain conformation and make for weaker intra- and inter-molecular interactions. Hasegawa et al. [29] synthesized a series of ester-linked diamines with different lengths and groups. They explained that incorporating substituents into the main chain can improve the solubility, and especially bulky substituents can influence water absorption behavior. Chen et al. [23] synthesized a bio-based PI with an ester-containing diamine monomer. The bio-based PI film showed excellent water absorption behavior (0.87–0.95). Han et al. [30] prepared a PI film using pyromellitic anhydride (PMDA), 4,4′-(hexafluoroisopropyl) phthalic anhydride (6FDA), 4,4′-diaminobiphenyl ether (ODA), and 2,2′-bis(trifluoromethyl)-4,4′-diaminobipheyl (TFMB) as the monomers and 1,3,5-tri(4-aminophenoxy) benzene (TAPOB) as the crosslinking agent. They studied the influence of molecular chain stiffness and flexibility on the micro-branched crosslink structure by both experimental and molecular dynamic simulation. Chen et al. [31] also synthesized 35 kinds of PI with ester, ether, and fluorine linkages. They mentioned that to achieve ultralow dielectric properties, the chemical structure of PI should not have polar groups, and thus ester moieties and hexafluoroisopropylidene groups were introduced into the main chain of PI. Introducing bulky groups into the PI main chain can improve the solubility and dielectric constant by increasing the intermolecular chain distance and reducing the CTC between the polymer chains [32]. Additionally, incorporating CF_3_ groups into the main chain can hinder chain packing and reduce the CTC due to the strong electron-withdrawing CF_3_ group and low polar -C-F bonds. This can also improve the solubility, optical properties, and dielectric properties of PI films [33]. The relationship between molecular structure and dielectric properties was investigated by Bei et al. [34]. They studied the use of PIs containing the same diphenylpyridine core structure but different side-chains with a varied number of benzene rings. They discussed the reason for the reduced dielectric constant value with the increasing number of benzene rings in the pendant group through the investigation of morphology, density, and water absorption properties of those polyimides. In addition, Zhang et al. [35] synthesized copolyimides from 6FDA, 2,2′-bis(trifluoromethyl)benzidine and 2,2-bis [4-(4-aminophenoxy)phenyl]hexafluoropropane. They reported that by adjusting the ratio of the two diamines, the chain density can be destroyed; thus, the hysteresis resistance of the dipole orientation can be weakened, and the dielectric loss can be as low as 0.00595, as well as exhibiting high thermal stability and mechanical properties. All these previous studies may help us to experimentally design new monomers for PIs having a low dielectric constant. Comprehensively, to achieve a colorless polyimide film possessing a low dielectric constant and low water absorption, designing a monomer that contains an ester group with bulky moiety seems to be an effective strategy. Additionally, synthesizing polyimide with a counter monomer that has a trifluoromethyl group can assist the goal of this research.

In this regard, a novel diamine monomer containing ester and phenyl moieties, 1,2-diphenylethane-1,2-diyl bis(4-aminobenzoate) (1,2-DPEDBA), was synthesized through a three-step reaction in this work. Using this diamine, a novel PI film was prepared with 6-FDA as a counter dianhydride through a typical two-step chemical imidization. Then, the chemical structure, optical properties, water barrier properties, thermal stability, and dielectric properties of the resulting PI film were investigated. We found that the resulting PI film has an outstanding optical transparency, low water absorption, and a very low dielectric constant.

## 2. Materials and Methods

### 2.1. Materials

Benzoin, sodium borohydride (NaBH_4_, ≥98.0%), 4-nitrobenzoyl chloride (4-NBC, 98%), methanol (MeOH, ≥99.8%), tetrahydrofuran (THF, ≥99.9%), N,N’-dimethylacetamide (DMAc, ≥99.8%), pyromellitic dianhydride (PMDA), and 4,4′-oxydianiline (ODA) were purchased from Sigma-Aldrich Korea (Seoul, Korea). Solvents such as N,N-dimethylformamide (DMF), dimethyl sulfoxide (DMSO), N-methyl pyrrolidone (NMP), chloroform (CHCl_3_), and ethanol were purchased from Junsei Chemical Co (Tokyo, Japan). As for 4,4-Diphthalic anhydride (6-FDA, ≥98.0%) and 10% palladium on carbon (Pd/C), they were purchased from Tokyo Chemical Industry Co. (TCI) (Tokyo, Japan). n-Hexane, ethyl acetate, and dichloromethane were purchased from Samchun Pure Chemicals (Pyeongtaek, Korea). Celite 545, which is a kind of diatomite whose main component is silica and is used to separate a catalyst from the reaction mixture, and γ-butyrolactone were purchased from Daejung (Siheung, Korea). The deionized water used in the experiments was purified by a Direct-Q^®^3 water purification system (EMD Millipore) (Busan, Korea). All purchased chemical reagents were used without further purification.

### 2.2. Synthesis of 1,2-Diphenylethane-1,2-diol

1,2-Diphenylethane-1,2-diol was synthesized according to the literature [36]. The synthesis of 1,2-Diphenylethane-1,2-diol was achieved by a reduction reaction using sodium borohydride as a catalyst. First, 5.0 g of benzoin was added to a 250 mL round bottom flask. Then, 80 mL of methanol was added as a solvent. After 30 min of stirring, 1.3 g of sodium borohydride (NaBH_4_) was slowly added to the flask in three portions. After stirring for 2 h, 50 mL of deionized water was poured into the flask. The mixture was separated using a separation funnel. The organic layer was collected and dehydrated by magnesium sulfate. After that, the solution was concentrated through a rotary evaporator. The final powder was obtained after drying at 80 °C overnight (yield: 95%, m.p.: 136.7 °C).

### 2.3. Synthesis of 1,2-Diphenylethane-1,2-diyl bis(4-Nitrobenzoate) (1,2-DPEDBN)

The synthesis of 1,2-Diphenylethane-1,2-diyl bis(4-nitrobenzoate) (1,2-DPEDBN) was achieved according to the literature [29] with slight modification (Figure 1). The synthesis of 1,2-Diphenylethane-1,2-diyl bis(4-nitrobenzoate) was achieved by reacting with 4-nitrobenzoyl chloride (4-NBC) in the presence of pyridine as an acid acceptor. First, 5.0 g of 1,2-diphenylethane-1,2-diol was added into a 500 mL round bottom flask sealed with a septum cap. Then, 25 mL of anhydrous tetrahydrofuran was added to the flask. The solution was stirred for 30 min and cooled at 0 °C. After that, 8.65 g of 4-nitrobenzoyl chloride was dissolved in 25 mL of THF in a separate round bottom flask under a nitrogen atmosphere. A 4-NBC solution was added dropwise with continuous stirring. Then, 11.29 mL of pyridine was slowly added into the flask. The mixture was stirred at room temperature for 12 h. The precipitate was collected by vacuum filtration and washed with deionized water and n-hexane. The final product was obtained after drying at 80 °C for 15 h (yield: 93%, m.p.: 235.1 °C).

### 2.4. Synthesis of 1,2-Diphenylethane-1,2-diyl bis(4-Aminobenzoate) (1,2-DPEDBA)

The overall design of the hydrogenation reaction refers to a previous work [37] (Figure 1). The preparation of 1,2-Diphenylethane-1,2-diyl bis(4-aminobenzoate) was achieved by a hydrogenation reaction using palladium on carbon (Pd/C) as a catalyst. First, 5.0 g of 1,2-DPEDBN and 0.5 g of palladium on carbon were added into a 250 mL round bottom flask under a nitrogen atmosphere. Then, 120 mL of anhydrous THF was added to the flask. After 30 min of stirring, hydrogen balloons were placed into the flask. The reaction was continuously stirred for 48 h. After that, the catalyst was filtered by Celite 525, and the filtrate was concentrated by a rotary evaporator. The final product was recrystallized by n-hexane and filtered. After vacuum drying at 80 °C for 15 h, 1,2-DPEDBA was obtained (yield: 85.4%, m.p.: 248.8 °C).

### 2.5. Preparation of 6FDA-DPEDBA PI and PMDA-ODA PI

As a precursor of PI, poly(amic acid) (PAA) was prepared by polycondensation polymerization with diamine and dianhydride. A solution of PAA with a 25 wt. % of solid content was prepared by a condensation reaction between 2.0 mmol of 1,2-DPEDBA (933 mg) and 2.0 mmol of 6-FDA under a nitrogen atmosphere. The solution was stirred for 24 h at room temperature. After that, 0.81 mL of pyridine and 1.89 mL of acetic anhydride were added dropwise into the solution. Under a nitrogen atmosphere, the solution was stirred for 24 h. After stirring was finished, the solution was slowly poured into 200 mL of methanol. The precipitate was filtered and washed with deionized water and methanol. The final product was obtained after drying at 80 °C for 24 h.

The 6FDA-DPEDBA film was prepared by dissolving 0.35 g of the PI powder into 2.78 mL of γ-butyrolactone (solid content of 10 wt.%). The solution was stirred until the solution became transparent and homogeneous. After that, the solution was cast onto a glass substrate and dried under a nitrogen atmosphere at 80 °C for 24 h. Figure 1 illustrates the synthetic route for 6FDA-DPEDBA PI as well as 1,2-DPEDBN and 1,2-DPEDBA.

For comparison, poly(pyromellitic dianhydride-co-4,4′-oxydianiline) (PMDA-ODA PI) was also synthesized. The PMDA-ODA PI synthetic route followed the literature [38]. A solution of PAA was prepared by polymerization and thermal imidization of diamine (ODA) and dianhydride (PMDA). First, equimolar amounts of ODA (2.0 mmol) and PMDA (2.0 mmol) were placed into 7.1 mL of DMAc. After stirring for 24 h under nitrogen at room temperature, PAA was poured onto a glass slide using a dropper and dried at 60 °C under a nitrogen atmosphere overnight. Next, the film was thermally imidized by the following process: 80 °C for 2 h, 120 °C for 1 h, 180 °C for 1 h, 250 °C for 0.5 h, and 300 °C for 0.5 h under a heating rate of 1 °C/min. The final film was obtained by immersing the slide glass in deionized water.

### 2.6. Measurements and Characterization

#### 2.6.1. Melting Point

The melting point (m.p.) of each organic compound was measured by differential scanning calorimeter (DSC) using Discovery DSC 25 (TA instruments) (New Castle, DE, USA) at a heating rate of 10 °C/min in the temperature range of 30–250 °C.

#### 2.6.2. Molecular Weight

The number-average molecular weight (M_n_) and weight-average molecular weight (M_w_) were analyzed using polystyrene standards by gel permeation chromatography (GPC) using a Waters 1515–2414 GPC (Milford, MA, USA) at 35 °C. Tetrahydrofuran (THF) was used as a solvent.

#### 2.6.3. X-ray Diffraction

Wide-angle X-ray diffraction (XRD, Bruker AXS) (Billerica, MA, USA) was conducted using Cu Kα radiation (λ = 1.5418 Å) at 40 kV and 40 mA at 2θ values of 5~80°.

#### 2.6.4. Thermogravimetric Analysis

The thermal stability of the PI film was characterized by thermogravimetric analysis (TGA, TGA Q500, TA instruments) (New Castle, DE, USA) at 30–800 °C with a heating rate of 10 °C/min under a nitrogen atmosphere.

#### 2.6.5. Water Absorption and Water Contact Angle

The water absorption behavior was tested by weighing the vacuum-dried PI film and soaking it in deionized water at room temperature for 24 h and calculated using Equation (1):(1)$$W_{A}=\left[\frac{W-W_{0}}{W_{0}}\right]\times 100$$
where *W*_0_ is the vacuum-dried PI film weight and *W* is the weight of the PI film wetted in deionized water for 24 h. The water contact angle of the PI film was analyzed by a contact angle analyzer (SEO Pheonix 300) (Suwon, Korea) according to the sessile drop method at room temperature using a 3 mL syringe with a needle of 0.127 mm in diameter. The water volume used in each measurement was 30 μL. Five readings of both static contact angles for each sample were taken and averaged to obtain a reliable value.

#### 2.6.6. Dielectric Constant

The dielectric properties of the PI film were analyzed by a broad frequency dielectric spectrometer (Concept 80, Novocontrol) (Montabaur, Germany) in the range of 10^0^–10^7^ Hz at 25 °C.

#### 2.6.7. FTIR Spectroscopy

The functional groups of the synthesized monomers and the novel PI film were characterized by Fourier transform infrared (FT-IR) spectroscopy using an FTIR-4100 (JASCO Co.) (Tokyo, Japan) in the range of 500–4000 cm^−1^ with a resolution of 2 cm^−1^.

#### 2.6.8. NMR Spectroscopy

The synthesized materials were also characterized by ^1^H-nuclear magnetic resonance (NMR) and ^13^C-NMR spectroscopy (Varian MR-400 spectrometer, Agilent Technologies) (Palo Alto, CA, USA) at a frequency of 400 MHz using deuterated dimethyl sulfoxide (DMSO-d_6_) as the NMR solvent.

#### 2.6.9. UV–Visible Spectroscopy

UV–visible optical transmission spectroscopy (UV–1650, Shimadzu) (Kyoto, Japan) was performed on the PI films in the wavelength range of 200–800 nm with a resolution of 0.5 nm and a scanning rate of 300 nm per min^−1^.

## 3. Results and Discussion

### 3.1. Synthesis and Characterization of the Diamine Monomer

The new ester- and phenyl-containing diamine monomer 1,2-DPEDBA was prepared through a total of three steps: ketone–alcohol reduction, esterification reaction with acyl halides, and hydrogenation reaction. The total synthesis route to 1,2-DPEDBA is shown in Figure 1. After these processes, the diamine monomer with ester and phenyl moieties was successfully synthesized. The functional groups and structures of 1,2-diphenylethane-1,2-diol, 1,2-DPEDBN, and 1,2-DPEDBA were analyzed by ^1^H-NMR, ^13^C-NMR, and FT-IR spectroscopy.

The ^1^H-NMR spectrum of 1,2-diphenylethane-1,2-diol is shown in Figure 1a, and all peaks are assigned to each proton in the chemical structure. In Figure 1a, the peaks at approximately 7.25~7.21 ppm (‘a, b, c’) were assigned to the aromatic protons. The peak at 5.20 ppm (‘d’) was assigned to the -CH group, while the peak at 4.55 ppm (‘e’) was assigned to the -OH group [33]. Figure 1b shows the ^1^H-NMR spectrum of 1,2-DPEDBN. The peaks at approximately 8.33 and 8.32 ppm (‘a, b’) were assigned to the benzene ring protons near the nitro group, while the peaks at 7.45 and 7.33 ppm (‘d, e, f’) were ascribed to protons in the phenyl group. The peak at 6.50 ppm (‘c’) was assigned to the -CH group. The ^1^H-NMR spectra of 1,2-DPEDBA are depicted in Figure 2a,b. In Figure 2a, the peaks at 7.62 and 6.54 ppm (‘a, b’) were assigned to the benzene ring protons near the amine group, while the peaks at approximately 7.29 ppm (‘d, e, f’) were assigned to the protons in the phenyl group [39]. The peaks at 6.24 ppm and 6.01 ppm (‘c, g’) were assigned to the -CH group and -NH_2_ group, respectively. In addition, Figure 2b shows the ^1^H-NMR spectrum of 1,2-DPEDBA with some drops of D_2_O. A comparison of Figure 2a,b indicates that only the peak at 6.01 ppm disappeared, meaning that this peak is assigned to the -NH_2_ proton [40].

The ^13^C-NMR spectra of 1,2-DPEDBN and DPEDBA are shown in Figure 3. All peaks in the figures are assigned to each carbon atom in the chemical structure. In Figure 3a, the peak at 162 ppm (‘e’) was assigned to the ester carbonyl (C=O) carbon atom, the peak at 150 ppm (‘a’) was assigned to the nitro group carbon (C-N) atom, and the peaks at approximately 120~135 ppm (‘b, c, d, g, h, i, j’) were assigned to the phenyl carbon atom. The peak at 77 ppm (‘f’) was assigned to ethyl carbon. In Figure 3b, the peak at approximately 160 ppm (‘e’) was assigned to the ester carbon atom, and the peak at approximately 150 ppm (‘a’) was assigned to amine carbon. The peak at approximately 76 ppm (‘f’) was assigned to ethyl carbon. The other peaks were assigned to phenyl carbon (‘b, c, d, g, h, i, j’) [41].

Furthermore, the functional groups of 1,2-DPEDBN and 1,2-DPEDBA were investigated by FT-IR spectroscopy. In Figure 4, for the 1,2-DPEDBN spectrum, the distinctive absorption bands at approximately 1528 and 1353 cm^−1^ were due to asymmetric and symmetric vibrations of the nitro groups (N=O), respectively. Additionally, the absorption bands at approximately 1731, 1267, and 1099 cm^−1^ were due to the vibration of the ester groups (C=O, C-O) [42]. After the hydrogenation reaction, distinctive absorption bands at approximately 3478, 3358, 1622, 1174, and 702 cm^−1^ newly appeared due to the primary amine groups (N-H, C-N).

### 3.2. Synthesis and Characterization of the 6FDA-DPEDBA PI

The new diamine monomer 1,2-DPEDBA was reacted with the counter dianhydride monomer 6-FDA to synthesize 6FDA-DPEDBA PI, as shown in Figure 1. Using typical two-step chemical imidization with pyridine and acetic anhydride (1:2), 6FDA-DPEDBA PI was prepared. The solid content of poly(amic acid) before chemical imidization was 25%, while the solid content of the PI solution was 10% using γ-butyrolactone as a solvent. In this study, the chemical imidization method was used rather than the thermal imidization method. The PI film prepared via chemical imidization may show good transparency because acetic anhydride can be an end-capping reagent for the reactive terminal group of the PI main chain, since the coloration problem of PI usually occurs at high temperatures [43].

The PI film was characterized by ^1^H-NMR spectrum, FT-IR spectrum, and GPC. The FT-IR spectrum of transparent and thin PI films was analyzed by using the attenuated total reflectance (ATR) method. As described in Figure 5a, the PI film showed typical imide group absorption bands at 1721 and 1370 cm^−1^ due to C-N asymmetric stretching and C=O asymmetrical stretching of the imide group, respectively. Furthermore, there are no N-H stretching vibration absorption bands at approximately 3200–3450 cm^−1^ or N-H bending vibration absorption bands at approximately 1550–1650 cm^−1^ due to the primary amine groups, implying that the chemical imidization process was successfully completed [44]. The ^1^H-NMR spectrum of PI is shown in Figure 5b. The peaks at approximately 8.0–8.3 ppm were assigned to the protons of dianhydride in the PI main chain, while the peaks at approximately 6.5–7.7 ppm were assigned to the protons of the diamine part in the PI main chain. Additionally, there are no more peaks at approximately 12 ppm for acid protons and approximately 5.5 ppm for amine protons, meaning that the chemical imidization process was successful. The average molecular weight of the PI was analyzed by using GPC. The number-average molecular weight of the PI was 1.06 × 10^4^ g mol^−1^, and the weight-average molecular weight of the PI was 2.37 × 10^4^ g mol^−1^. Additionally, the polydispersity index of the PI was 2.19. Overall, it can be said that our novel diamine reacted with the counter dianhydride and produced PI with a sufficient molecular weight.

### 3.3. Solubility of PI

The solubility of the synthesized PI was tested by dissolving 10 mg of the PI powder into 1 mL of each organic solvent at room temperature in a test tube. As shown in Table 1, the synthesized PI shows excellent solubility not only in common polar solvents with high boiling points, such as DMAc and DMF, but it is also soluble in common low-boiling-point polar solvents, such as chloroform (CHCl_3_) and dichloromethane (CH_2_Cl_2_), at room temperature. This can be explained by several reasons. First, introducing a bulky phenyl group into the main chain of PI can increase the intermolecular space and free volume between the polymer chains. Additionally, introducing a bulky group can decrease the chain packing density between the polymer chains and increase polymer flexibility [45]. Second, introducing an ester moiety into the PI main chain can hinder the CTC between the dianhydride and diamine moieties and decrease intermolecular interactions. This can increase the overall polymer chain flexibility and chain segment mobility and decrease the chain packing density [46]. Last, using 6-FDA as a counter dianhydride containing CF_3_ groups can significantly increase polymer solubility. It was reported that bulky CF_3_ groups can increase interchain space and hinder chain packing [47,48]. Additionally, CF_3_ groups can reduce intermolecular forces due to the low polarizability of the C-F group. Overall, introducing ester and phenyl groups into the diamine monomer can effectively hinder intermolecular forces and easily penetrate the solvent molecule into the polymer matrix. Additionally, using 6-FDA as a counter dianhydride can escalate this phenomenon.

### 3.4. X-ray Diffraction of the 6FDA-DPEDBA PI Film

The microscopic crystallinity of the PI film was analyzed by using XRD, and the result is described in Figure 6. The diffractogram shows a 2θ value from 10° to 80°. The diffractogram of the PI film displayed wide diffraction peaks at approximately 20° and overall broad peaks, meaning that the resulting PI film was amorphous. This result is due to the fact that introducing a bulky phenyl group and a trifluoromethyl group disturbs the intermolecular packing and increases the intermolecular space and free volume. This can hinder the crystalline order of the polymer matrix. For these reasons, the PI crystallinity was deteriorated by hindering the intermolecular interactions and intermolecular chain packing [49].

### 3.5. Properties of the 6FDA-DPEDBA PI Film

#### 3.5.1. Thermal Stability of the 6FDA-DPEDBA PI Film

The thermal stability of the 6FDA-DPEDBA PI film was analyzed by using TGA under a nitrogen atmosphere with a heating rate of 10 °C/min. The TGA curve is shown in Figure 7a. The PI film showed a 5% weight loss temperature at 360 °C and a 10% weight loss temperature at 375 °C. These results imply some deterioration of the thermal stability of the PI in comparison with the PMDA-ODA PI film (T^5^_d_ = 564 °C). Due to the fact that the diphenylethane group was introduced into the main chain of the PI film, the aromatic conjugation structure in the PI main chain was discontinued. Even though the bulky diphenylethane group can improve solubility, optical transparency, and dielectric properties, it can also decrease the interaction between the polymer chains, and the existence of a carbon single bond (C-C) in the polymer main chain may cause a break at a higher temperature [50]. However, this kind of deterioration in thermal stability may be easily overcome in a future study by the hybridization with various kinds of inorganic materials, such as silica, polyhedral oligomeric silsesquioxane (POSS), graphene, MXene, etc. [51,52,53,54].

#### 3.5.2. Optical Properties of the 6FDA-DPEDBA PI Film

The optical properties of the 6FDA-DPEDBA PI film were analyzed by UV–Vis spectrometry over a range of 200–800 nm. The thickness of the 6FDA-DPEDBA PI film was 40 μm, while that of the PMDA-ODA PI film was 38 μm. The results are described in Figure 7b and Table 2. The PI film showed colorless and outstanding transparency in the visible light region from 400 nm to 760 nm. The cutoff wavelength of the PI film was 365 nm, the average transmittance in the visible light region (T_vis_) was 87.1%, and the transmittance at 450 nm (T_450_) was 85.57%. The typical fully aromatic PMDA-ODA PI film showed a yellow-dark brown color due to the chemical structure of the PI main chain. The optical images of both the 6FDA-DPEDBA PI and PMDA-ODA PI film were displayed in the inset of Figure 7b for reference. The intra- and intermolecular CTC between electron accepting dianhydride and electron donating diamine is the main reason for the coloration [47]. The presence of a bulky diphenyletane group increases the intermolecular chain space and free volume [55]. Therefore, the CTC between other molecules is effectively hindered. Additionally, introducing an ester group into the PI main chain can easily decrease the imide group content, which can cause a CTC. Furthermore, the counter dianhydride 6-FDA has low polarizability and bulky trifluoromethyl groups [56]. Additionally, we can calculate the band gap (Δε) between the highest occupied molecular orbital (HOMO) and the lowest unoccupied molecular orbital (LUMO) from the UV–Vis absorbance data. Practically, Δε can be estimated from the wavelength at half of the excitonic peak [57,58,59]. The calculated Δε of 6FDA-DPEDBA PI was ~3.20 eV, which is higher than that of PMDA-ODA PI (~2.45 eV). This result implies that the strong interaction due to the conjugation effect between polymer chains is effectively reduced for the molecular structure of the 6FDA-DPEDBA PI film.

Overall, the strategy of introducing ester and bulky phenyl groups into the diamine monomer, using 6-FDA as a counter dianhydride, can reduce the coloration of the film, increase the energy band gap between the HOMO and LUMO of the polymer, and hinder the CTC in the PI matrix [41]

#### 3.5.3. Water Absorption and Water Contact Angle of the 6FDA-DPEDBA PI Film

The water absorption behavior was tested by weighing a vacuum-dried 6FDA-DPEDBA PI film and soaking it in deionized water at room temperature for 24 h. Then, it was calculated using Equation (1), as described in Section 2.6. In this study, *W* of the resulting PI film was 0.1086 g, and *W*_0_ was 0.1067 g. According to Equation (1), the water absorption of the PI film is of 1.78%, which is lower than that of PMDA-ODA PI film, i.e., conventional PI films such as KAPTON PI [60]. Introducing a bulky phenyl group increases intermolecular spaces and free volume and loosens the chain packing of the PI backbone, providing enough space for entrapping water molecules into the polymer matrix. Even though this phenomenon occurs, introducing an ester group into the polymer backbone ensures that the resulting PI film maintains a lower water absorption value than commercial PI. The water absorption is roughly related to the imide group content [29]. The highly polarized imide group and carbonyl group in the ester moiety can participate in the water absorption mechanism. However, the contribution of the carbonyl group is not more significant than reducing the imide group by introducing an ester moiety into the PI backbone.

The water contact angle of the PI film was analyzed by a contact angle analyzer. The average contact angle of the PI film was 90.26°. It was stated that if the water contact angle of the film is higher than 75°, then the water barrier properties of the film are relatively strong and hydrophobic [61].

Overall, the PI film based on a novel diamine containing ester and phenyl groups exhibits improved transparency and solubility by increasing the intermolecular chain space while maintaining the water barrier properties. Considering that the barrier against humidity is important for dielectric materials, the resultant PI film shows promising potential.

#### 3.5.4. Dielectric Constant of PI Films

The dielectric constant of the resultant 6FDA-DPEDBA PI film was analyzed by a broad frequency dielectric spectrometer and is described in Figure 7c. The dielectric constant of the 6FDA-DPEDBA PI film was measured at a frequency of 10^0^ to 10^7^ Hz at 25 °C. The resultant PI film showed a very low dielectric constant of 2.17 at 1 MHz, while the PMDA-ODA PI film showed a dielectric constant of 3.59 at 1 MHz. This outstanding dielectric constant of the PI film was due to the low imide group content due to the introduction of an ester group and a bulky phenyl group in the PI backbone as well as the presence of a strong electron-withdrawing trifluoromethyl group. Due to the aforementioned reasons, the 6FDA-DPEDBA PI film has a twisted and flexible chemical structure compared to the rigid, tightly packed PMDA-ODA PI film. The bulky phenyl group increases the distance between polymer chains and reduces intermolecular CTC interactions. Additionally, the ester group could inhibit the interaction between the diamine moiety and dianhydride moiety [62]. Moreover, introducing the diphenylethane group can destroy the conjugated structure of the PI main chain and hinder the free flow of charges. Furthermore, the strong electron-withdrawing nature of the trifluoromethyl group containing 6-FDA effectively reduces the dielectric constant. According to the Clausius–Mossotti equation described in Equation (2), the chain polarizability and free volume of polymers are the main factors for the dielectric constant (*D_k_*) [63].
(2)$$D_{k}=\left(1+\frac{2P}{V}\right)/\left(1-\frac{P}{V}\right)\;$$

Following the above equation, the strategy of making PI with ester- and phenyl-containing diamine and 6-FDA not only increases the molar volume (*V*) but also decreases the molar polarity (*P*) [27].

Considering that the 6FDA-DPEDBA PI film has excellent dielectric properties, optical transparency, and water absorption behavior, it may be expected that this novel polyimide film be a promising candidate for a next-generation communication device such as organic field-effect transistors (OFETs), thin film transistors (TFTs), and optical electronic devices such as active-matrix organic light emitting display devices (AMOLEDs) [64,65,66]. Furthermore, our findings may give insights for a guideline to further reduce the dielectric constant to meet the application requirements of modified PI (MPI) in 5G equipment, by extending our findings in this work, so that any as-synthesized polyimides can put forward good comprehensive properties with an easy strategy for interlayer dielectric (ILD) materials in the high-frequency communication of 5G [35,67].

## 4. Conclusions

In this study, a novel diamine monomer containing ester and phenyl moieties was synthesized through a three-step reaction and reacted with 6-FDA as a counter dianhydride by the chemical imidization method. For comparison, poly(pyromellitic dianhydride-*co*-4,4′-oxydianiline) (PMDA-ODA PI) was also synthesized via thermal imidization. The resultant 6FDA-DPEDBA PI film has a high average molecular weight. The 6FDA-DPEDBA PI film showed good solubility in various polar organic solvents, such as DMAc, DMSO, CH_2_Cl_2_, and CHCl_3_. Additionally, the resultant PI film showed outstanding transparency (T_vis_ = 87.1% of transmittance of visible light), low water absorption (W_A_ = 1.78%), and a low dielectric constant (2.17 at 1 MHz). In summary, the novel PI film showed much better optical transparency, lower moisture absorption, and a lower dielectric constant as well as better solubility than the PMDA-ODA PI film, which is insoluble in any solvent, although its thermal stability is not better than that of PMDA-ODA PI. Overall, the strategy of introducing an ester moiety and bulky phenyl group into the PI backbone can achieve a low dielectric constant and low water absorption while maintaining outstanding optical transparency and solubility. Thus, the developed PI with the novel diamine in this work can be a promising candidate as a suitable material for next-generation communication devices, dielectrics, optical electronic devices, etc.

## Data Availability

Not applicable.

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
