# Peer review of "A Novel Diamine Containing Ester and Diphenylethane Groups for Colorless Polyimide with a Low Dielectric Constant and Low Water Absorption"

_polymers, 2022, doi:10.3390/polym14214504_

Round 1

Reviewer 1 Report

The manuscript entitled as A Novel Diamine Containing Ester and Diphenylethane Groups for Colorless Polyimide with a Low Dielectric Constant and Low Water Absorption” investigate development of soluble form of polyimides possessing high optical transparency and low dielectric constant and low water absorption for applications of optoelectronic devices.

The comments are addressed below:

1.      Introduction section can be extended by adding more literature examples.

2.      Explain ‘Celite 525’ in line 128 and ‘m.p.’ in line 107 and others

3.      When you mention the details of supplied materials or used devices, please be constant at adding the city and country information of the manufacturer/supplier in the parenthesis.

4.      Lines between 185-196 and ‘Scheme 1’ at Page 5 explain the general synthesis method. Therefore, it is better to go in the methodology section not in the Results and discussion. This explanation can be placed under a new title as e.g. 2.2 Synthesis procedure following the ‘Materials’ section. The specific synthesis steps can be given as subtitles, such as ‘2.2.1. Synthesis of 1,2-diphenylethane-1,2-diol ‘.

5.      Each characterization techniques, such as FTIR, NMR, UV in section 2.6, can be explained separately with different titles.

Author Response

Dear Reviewer,

Thank you very much for your valuable comments regarding our manuscript entitled “A Novel Diamine Containing Ester and Diphenylethane Groups for Colorless Polyimide with a Low Dielectric Constant and Low Water Absorption”. We are truly grateful to the comments and suggestions from all of you. The manuscript has been carefully revised, and the point-to-point answers to the comments and suggestions are listed as below, for you reference.

The manuscript entitled as “A Novel Diamine Containing Ester and Diphenylethane Groups for Colorless Polyimide with a Low Dielectric Constant and Low Water Absorption” investigate development of soluble form of polyimides possessing high optical transparency and low dielectric constant and low water absorption for applications of optoelectronic devices.

The comments are addressed below:

Q1; Introduction section can be extended by adding more literature examples.

Answer 1: Thank you for your valuable suggestion. According to the reviewer’s suggestion, we added some more description of previous key findings on related subject with referring to several new literatures [28,30,34,35] as well as already-provided references [29,31-33] in the introduction section (L66-L79; L88-L104).

Q2: Explain ‘Celite 525’ in line 128 and ‘m.p.’ in line 107 and others

Answer 2: Firstly, thanks for this comment from the reviewer. The right name is ‘Celite 545’ we made a mistake. Thank you for pointing it out. The Celite 545 is a kind of diatomite whose main component is silica. It is generally used to separate a catalyst from the reaction mixture. We added the explanation on Celite 545 in L124-L125.

The meaning of ‘m.p.’ is the melting point of each organic compound. It was measured from Differential Scanning Calorimeter(DSC) using Discovery DSC 25(TA instruments) at a heating rate of 10℃/ min in the temperature range of 30-250℃. This information was added in the measurement and characterization section (L193-L196).

Q3; When you mention the details of supplied materials or used devices, please be constant at adding the city and country information of the manufacturer/supplier in the parenthesis.

Answer 3: Thank you very much for your useful comments. According to the reviewer’s suggestion. we added suppliers’ information for materials and devices by adding the city and country name in Pages 5-6.

Q4; Lines between 185-196 and ‘Scheme 1’ at Page 5 explain the general synthesis method. Therefore, it is better to go in the methodology section not in the Results and discussion. This explanation can be placed under a new title as e.g. 2.2 Synthesis procedure following the ‘Materials’ section. The specific synthesis steps can be given as subtitles, such as ‘2.2.1. Synthesis of 1,2-diphenylethane-1,2-diol ‘.

Answer 4: Firstly, thanks for this comment from the reviewer. The general synthesis method was already mentioned in the materials and methods section. Therefore, we removed detailed synthesis procedure according to your kind advice. Additionally, we moved the Scheme 1 to the synthesis procedure section (Pages 3-5). Instead, we left just the outline of the simple reaction pathway in 3.1. section (L 246-252).

Q5; Each characterization techniques, such as FTIR, NMR, UV in section 2.6, can be explained separately with different titles.

Answer 5: Thank you for your valuable advice. According to the reviewer’s suggestion, we categorized the 2.6 measurement and characterization section into 2.6.1. Melting point through 2.6.9. UV-vis spectroscopy.

We thank you very much for your valuable comments and suggestions. Our revision is highlighted in red in this revised manuscript. We did our best to incorporate your valuable comments in this revised manuscript. We believe the quality of this revised manuscript would have been significantly improved. We hope that our revision could have been done successfully.

Reviewer 2 Report

The research described in this manuscript is interesting, it gives an aletrnative of material develop according to the tittle and objective of the manuscript. It is suggested that the experimental design should be mentiones and it should also be included the stadistically procedure that was folloed. The method escribed are sutable to obtain results according to the objetive that is not define as that but it can be infered that was it is written on the last paragraph of the manuscript. The manuscriptcan be improve by including what it is suggested. The figures are in color and can be read really easily. 

Author Response

Dear Reviewer,

Thank you very much for your valuable comments regarding our manuscript entitled “A Novel Diamine Containing Ester and Diphenylethane Groups for Colorless Polyimide with a Low Dielectric Constant and Low Water Absorption”. We are truly grateful to the comments and suggestions from all of you. The manuscript has been carefully revised, and the point-to-point answers to the comments and suggestions are listed as below, for you reference.

Q; The research described in this manuscript is interesting, it gives an aletrnative of material develop according to the tittle and objective of the manuscript. It is suggested that the experimental design should be mentioned and it should also be included the statistically procedure that was followed. The method described are suitable to obtain results according to the objective that is not defined as that but it can be inferred that was it is written on the last paragraph of the manuscript. The manuscript can be improved by including what it is suggested. The figures are in color and can be read really easily. 

Answer; Many thanks for your valuable comments. According to your kind advice, we added more sentences on the approach of our experimental design of the present work with referring to several literatures in the Introduction (L88-L100). Also in the second last paragraph of the Introduction, we highlighted the objective of the present work to more degree(L100-L104). Also we added a few more sentences on the further application of the present findings for 5G telecommunication and other optoelectronic devices, etc. in the end of the Discussion section with referring to a few new literatures ([35,65-68])(L455-L464).

We thank you very much for your valuable comments and suggestions. Our revision is highlighted in red in this revised manuscript. We did our best to incorporate your valuable comments in this revised manuscript. We believe the quality of this revised manuscript would have been significantly improved. We hope that our revision could have been done successfully.

Reviewer 3 Report

Journal name: Polymers

Manuscript number: 1973691

Title: A Novel Diamine Containing Ester and Diphenylethane Groups for Colorless Polyimide with a Low Dielectric Constant and Low Water Absorption

Through the manuscript, the reviewer has some comments as below:

1. Originality: This work contributes to the synthesis of a diamine monomer which is used to fabricate a novel polyimide with a low dielectric constant and low water absorption.

2. Scientific quality: This manuscript provides a good scientific quality.

3. Relevance to the field(s) of this Journal: This manuscript matches perfectly the field covered by the journal.

4. General comment: Overall, the manuscript clearly describes the method to fabricate a monomer for a polyimide which possesses remarkable optical and electrical properties. With the future work to improve the thermal stability, this new polymer will be a promising candidate as a suitable material for next-communication devices, optical electronics devices, etc.

5. Abstract: This abstract clearly provides information related to the purpose, methodology, and brief results of the work. The keywords are valuable and suitable for this research.

6. Introduction: The introduction has well described the background information and research objectives. The research questions are also proposed scientifically.

7. Literature review: The literature review has well provided the related, updated research to the work.

8. Methodology: From the concept to the implementation process (synthesis of novel diamine, create novel PI, determine the physical properties of new material), the method of research was clearly and systematically described.

9. Results: The results were clearly presented and analyzed with remarkable evidence. The used methods and tools for determination of optical, di-electrical, thermal properties of the new PI are suitable.

The reviewer has a question on the thickness of 6FDA-DPEDBA layer which was used to measure the transmittance in subsection 3.5.2 (line 317-322). Please clearify it.

10. Discussions: The questions provided for the results were scientific and necessary. The explanations were reasonable.

11. Conclusions: The conclusion has well summarized the results of the work and also proposed the future work for improving the properties of new synthesized PI.

12. References/Bibliography: The references were well presented in an appropriate format. They were matched with the citations in the manuscript.

13. Figures: The figures were well presented with appropriate size. It should be better if the author can improve the graphic resolution because some figures contain small digits and they are difficult to observe.

14. Tables: Tables were well presented in appropriate size and format.

15. Reviewer’s decision comment: The manuscript is well edited, the content is reasonably scientific and novel. The reviewer proposes that this manuscript is acceptable to be published after minor revision.

Author Response

Dear Reviewer,

Thank you very much for your valuable comments regarding our manuscript entitled “A Novel Diamine Containing Ester and Diphenylethane Groups for Colorless Polyimide with a Low Dielectric Constant and Low Water Absorption”. We are truly grateful to the comments and suggestions from all of you. The manuscript has been carefully revised, and the point-to-point answers to the comments and suggestions are listed as below, for you reference.

Through the manuscript, the reviewer has some comments as below:

  1. Originality:This work contributes to the synthesis of a diamine monomer which is used to fabricate a novel polyimide with a low dielectric constant and low water absorption.
  2. Scientific quality:This manuscript provides a good scientific quality.
  3. Relevance to the field(s) of this Journal:This manuscript matches perfectly the field covered by the journal.
  4. General comment:Overall, the manuscript clearly describes the method to fabricate a monomer for a polyimide which possesses remarkable optical and electrical properties. With the future work to improve the thermal stability, this new polymer will be a promising candidate as a suitable material for next-communication devices, optical electronics devices, etc.
  5. Abstract:This abstract clearly provides information related to the purpose, methodology, and brief results of the work. The keywords are valuable and suitable for this research.
  6. Introduction:The introduction has well described the background information and research objectives. The research questions are also proposed scientifically.
  7. Literature review:The literature review has well provided the related, updated research to the work.
  8. Methodology:From the concept to the implementation process (synthesis of novel diamine, create novel PI, determine the physical properties of new material), the method of research was clearly and systematically described.
  9. Results:The results were clearly presented and analyzed with remarkable evidence. The used methods and tools for determination of optical, di-electrical, thermal properties of the new PI are suitable.

Answer; Many thanks for your encouraging comments.

Q1; The reviewer has a question on the thickness of 6FDA-DPEDBA layer which was used to measure the transmittance in subsection 3.5.2 (line 317-322). Please clarify it.

Answer 1: Firstly, thank you for your comment. The thickness of 6FDA-DPEDBA film was in the Table 2. However, considering your valuable comment, we added the short explanation about the thickness in the section 3.5.2.(L374-L375)

  1. Discussions:The questions provided for the results were scientific and necessary. The explanations were reasonable.
  2. Conclusions:The conclusion has well summarized the results of the work and also proposed the future work for improving the properties of new synthesized PI.
  3. References/Bibliography:The references were well presented in an appropriate format. They were matched with the citations in the manuscript.

-Many thanks for your encouraging comments.

  1. Figures:The figures were well presented with appropriate size. It should be better if the author can improve the graphic resolution because some figures contain small digits and they are difficult to observe.

Answer: Thank you very much for your valuable suggestion. The advice is very useful to improve the quality of our manuscript. We adjusted the images of 1H-NMR and 13C-NMR with better resolutions. Also we enlarged the fonts of letters in all figures for easy observation including Scheme 1 and Figure 7.

  1. Tables:Tables were well presented in appropriate size and format.
  2. Reviewer’s decision comment:The manuscript is well edited, the content is reasonably scientific and novel. The reviewer proposes that this manuscript is acceptable to be published after minor revision.

  • Many thanks for your encouraging comments.

We thank you very much for your valuable comments and suggestions. Our revision is highlighted in red in this revised manuscript. We did our best to incorporate your valuable comments in this revised manuscript. We believe the quality of this revised manuscript would have been significantly improved. We hope that our revision could have been done successfully.

  •  
